# Myonuclear accretion is a determinant of exercise-induced remodeling in skeletal muscle

Qingnian Goh[1], Taejeong Song[2], Michael J Petrany[1], Alyssa AW Cramer[1], Chengyi Sun[1], Sakthivel Sadayappan[2], Se-Jin Lee[3,4], Douglas P Millay[1,5]*

[1]Division of Molecular Cardiovascular Biology, Cincinnati Children's Hospital Medical Center, Cincinnati, United States; [2]Department of Internal Medicine, Division of Cardiovascular Health and Disease, University of Cincinnati College of Medicine, Cincinnati, United States; [3]The Jackson Laboratory, Farmington, United States; [4]Department of Genetics and Genome Sciences, University of Connecticut School of Medicine, Farmington, United States; [5]Department of Pediatrics, University of Cincinnati College of Medicine, Cincinnati, United States

**Abstract** Skeletal muscle adapts to external stimuli such as increased work. Muscle progenitors (MPs) control muscle repair due to severe damage, but the role of MP fusion and associated myonuclear accretion during exercise are unclear. While we previously demonstrated that MP fusion is required for growth using a supra-physiological model (Goh and Millay, 2017), questions remained about the need for myonuclear accrual during muscle adaptation in a physiological setting. Here, we developed an 8 week high-intensity interval training (HIIT) protocol and assessed the importance of MP fusion. In 8 month-old mice, HIIT led to progressive myonuclear accretion throughout the protocol, and functional muscle hypertrophy. Abrogation of MP fusion at the onset of HIIT resulted in exercise intolerance and fibrosis. In contrast, ablation of MP fusion 4 weeks into HIIT, preserved exercise tolerance but attenuated hypertrophy. We conclude that myonuclear accretion is required for different facets of exercise-induced adaptive responses, impacting both muscle repair and hypertrophic growth.
DOI: https://doi.org/10.7554/eLife.44876.001

*For correspondence:
douglas.millay@cchmc.org

Competing interests: The authors declare that no competing interests exist.

## Introduction

Skeletal muscle is a unique tissue that comprises 40% of our total body mass and is essential for breathing, movement, and metabolism. In contrast to the majority of other cell types, a single skeletal muscle fiber, which is formed from the fusion of precursor cells during development, contains hundreds of nuclei (*Deng et al., 2017*). Muscle is a remarkably plastic tissue in the adult with the ability to regenerate due to the presence of *bona fide* muscle stem cells (MuSCs or also referred to as satellite cells), and also respond to various stimuli by altering its contractile and metabolic properties (*Egan and Zierath, 2013*; *Hoppeler, 2016*). Proper regulation of skeletal muscle involves adaptations in the myofiber and an ability for MuSCs to activate and differentiate into muscle progenitors (MPs), which fuse to myofibers thereby increasing myonuclear number. While the role for MP fusion in response to major damage such as cardiotoxin-injury is well-recognized, its function during exercise-induced adaptations continues to be debated.

The adaptive response of skeletal muscle to exercise is diverse and includes an alteration in structure, metabolism, and function. The specific adaptation depends on the exercise stimulus, where resistance training induces an increase of myofiber cross-sectional area (CSA) to deal with increased loads and endurance training elicits greater metabolic changes allowing muscle to become more

fatigue resistant. There is consensus that MuSCs are indeed activated in humans after exercise, however whether MuSC activation and downstream myonuclear accretion is required for adaptive hypertrophy is not fully understood (*Snijders et al., 2015*; *Murach et al., 2018a*; *Bamman et al., 2018*). The requirement of myonuclear accretion for hypertrophy may depend on muscle fiber type as growth of 18% in slow, type I myofibers is associated with an increase of myonuclei (*Snijders et al., 2016*; *Bellamy et al., 2014*). In contrast, an increase in myonuclear content of fast, type II myofibers is only observed after 26–40% hypertrophy (*Snijders et al., 2016*; *Kadi et al., 2004*; *Petrella et al., 2006*; *Stec et al., 2016*). Taken together, while muscle adapts to exercise in various ways, the role of MP fusion and associated myonuclear accretion for hypertrophy is not clear.

While informative, the majority of human studies on the response of MuSCs to exercise are correlational in nature and biopsies reflect only a small portion of the entire muscle, therefore model organisms such as the mouse have also been utilized to shed light on MuSC-dependent exercise adaptations. Muscle overload by synergist ablation is the most often used stimulus to drive hypertrophy in the mouse, however it is an invasive procedure where a synergist muscle (e.g., gastrocnemius) is surgically removed leaving a smaller muscle (e.g., plantaris) to do all of the required work to ambulate the limb (*Kirby et al., 2016a*). While synergist ablation induces significant hypertrophy, there are questions about its physiological relevance due to its invasive nature and the potential to damage the muscle, and the magnitude of insult is far above anything experienced by muscle undergoing normal exercise. Recent work using models of MuSC ablation through diptheria toxin (DTA) has shown differing results as to whether they are required for hypertrophy in response to synergist ablation (*Egner et al., 2016*; *Egner et al., 2017*; *McCarthy et al., 2017*; *McCarthy et al., 2011*). Current ideas include that MP fusion is required for synergist ablation at young ages (below 4 months of age) but not at older timepoints (*Murach et al., 2017a*). Through a complementary approach, we abrogated MP fusion by genetically deleting Myomaker (*Mymk*), a muscle-specific factor required for fusion, and found that synergist ablation-induced hypertrophy was significantly blunted (*Goh and Millay, 2017*). These results are consistent with Egner et al., that the presence of MuSCs are required for growth induced by synergist ablation, but questions still exist about whether there is an age dependence of MP fusion for hypertrophy and if they are required in a physiologically relevant system. While voluntary wheel running is a standard exercise stimulus for the mouse and ablation of MuSCs by DTA have minimal impact on adaptations to voluntary wheel running (*Jackson et al., 2015*; *Murach et al., 2017b*), this exercise stimulus does not lead to significant myofiber hypertrophy (*De Lisio and Farup, 2017*). A progressive weighted wheel running protocol was recently reported to increase myofiber size but it is not understood if this exercise regimen is associated with improved muscle function or if MP fusion is required (*Dungan et al., 2019*). Thus, it is not known if MP fusion is necessary for adaptive hypertrophy in a physiologically-relevant exercise protocol.

In addition to the lack of a physiological hypertrophy-inducing protocol for mice, another challenge in elucidating the role of MP-dependent myonuclear accretion during exercise-induced adaptations is that changes in muscle are dynamic. Indeed, initiation of an exercise regimen leads to mechanical damage of the muscle characterized by alterations of the cytoskeleton and structural proteins (*Damas et al., 2018*). Because the classical stimulus activating MuSCs is muscle damage it is not surprising they are stimulated in this initial stage of exercise. It has thus been proposed that myonuclear accretion during exercise may occur as an early injury response and not specifically to drive hypertrophy (*Murach et al., 2018a*). Moreover, it is not understood if the increase in myonuclei resulting from accrual during the early response to exercise are sufficient for myofiber growth that occurs at later stages. Overall, a major obstacle in elucidating a causative relationship between MP fusion and myofiber growth has been the lack of a model that uncouples the addition of myonuclei for repair from accretion associated with hypertrophy.

We therefore sought to develop an exercise protocol that leads to muscle hypertrophy, thereby allowing interrogation of the requirement of MP fusion during the process. We utilized a high-intensity interval treadmill protocol (HIIT) that led to progressive myonuclear accretion, consistent hypertrophy of multiple muscles, and improved functional output. Through abrogation of MP fusion by deleting Myomaker in MPs, we demonstrate that myonuclear accrual is essential for normal adaptive

remodeling in response to exercise. Most notably, our data indicate that accretion of myonuclei supports temporally distinct adaptive outcomes, which allows us, for the first time, to parse apart myonuclear accrual occurring in response to muscle damage and repair from growth.

## Results

### A high-intensity interval training (HIIT) exercise protocol elicits an increase in muscle weight and strength

A hypertrophy-inducing exercise protocol for mice is lacking (*Cholewa et al., 2014*), although recently a high interval training protocol (HIIT) for aged mice was reported (*Seldeen et al., 2018*). To determine the role of MP fusogenicity for physiological hypertrophy, we modified this protocol for non-aged adult mice. The aged HIIT protocol lasted 16 weeks and we hypothesized hypertrophy could be maximized in 8 weeks if the amount of work accomplished over 16 weeks could be condensed over 8 weeks. We designed an exercise regimen to increase the amount of work during each week of the protocol by training mice three times per week and either increasing the angle of incline or speed of the treadmill (*Figure 1A*). After 1 week of acclimation to the treadmill, 8-month-old wild-type (WT) mice were subjected to a treadmill speed of 15–16 meters/minute with an incline of 10°. Speed remained constant during the initial 4 weeks of the protocol, but incline was increased by 5° each week. For the final 4 weeks, incline was maintained at 25° but speed was progressively increased. Analysis of muscle weights revealed a significant increase of multiple muscle groups in HIIT-trained mice compared to sedentary mice (*Figure 1B*). We next performed in vivo functional analysis to assess if HIIT resulted in an increase in muscle strength. The hindlimb plantar flexor muscles were stimulated at multiple frequencies and force was measured throughout each of these contractions, generating a force-frequency curve (*Figure 1C*). Peak isometric tetanic force of the hindlimb plantar flexor was increased after HIIT compared to sedentary mice (*Figure 1D*). The peak isometric tetanic force curve was then analyzed to calculate the rates by which the muscles contracted and relaxed. We observed an increase in rate of activation (*Figure 1E*) but no significant effect on rate of relaxation (*Figure 1F*), indicating that HIIT-trained muscle has the capability to generate peak tetanic force more quickly. Finally, calculation of impulse (area under the peak isometric tetanic force curve) revealed an increase after HIIT showing that those muscles can sustain higher force over a single contraction (*Figure 1G*). Overall, HIIT leads to an increase in muscle size and strength suggesting that it could represent a novel protocol to investigate exercise-induced adaptations in the mouse.

### HIIT induces a progressive increase in myonuclei and hypertrophy

We next assessed the magnitude and timing of myonuclear accretion at various stages of HIIT. We stained nuclei of isolated myofibers from the extensor digitorum longus (EDL) from sedentary mice and mice subjected to 2, 4, and 8 weeks of HIIT. Qualitatively, we observed an increase of myonuclei at each stage of the protocol (*Figure 2A*). Quantification of the number of nuclei per myofiber revealed 227 ± 15 myonuclei from sedentary muscle, whereas myofibers after 2 weeks of HIIT contained 253 ± 11 myonuclei (*Figure 2B*). After 4 weeks of HIIT myofibers contained 281 ± 9 and at 8 weeks there were 306 ± 12 myonuclei (*Figure 2B*). To complement the analysis of myonuclei in isolated EDL myofibers, we also assessed myonuclear number on cryosections from quadriceps by evaluating the number of nuclei within a dystrophin-stained myofiber. This analysis also revealed an increase in myonuclei (*Figure 2—figure supplement 1A*), consistent with the analysis in EDL myofibers. Moreover, analysis of the relative increase in myonuclei in the quadriceps and EDL showed that the different methods to quantify nuclear number yield similar results (*Figure 2—figure supplement 1B*). These data demonstrate that there is a continuous increase in myonuclei throughout the exercise regimen.

An increase in myofiber size is a normal adaptation to increased workload. We assessed myofiber hypertrophy at multiple time points after HIIT in both the gastrocnemius and quadriceps. Analysis of muscle sections revealed an increase of myofiber size after the entire 8 week HIIT protocol, with minimal gains in size at 2 or 4 weeks post-HIIT (*Figure 2C*). Quantification of myofiber cross-sectional area (CSA) indeed showed a significant increase at 8 weeks but not at prior time points (*Figure 2D*). Thus, muscle responds to HIIT by increasing cell size and this hypertrophy occurs between 4 and 8 weeks of the protocol, which is delayed compared to the initiation of myonuclear addition.

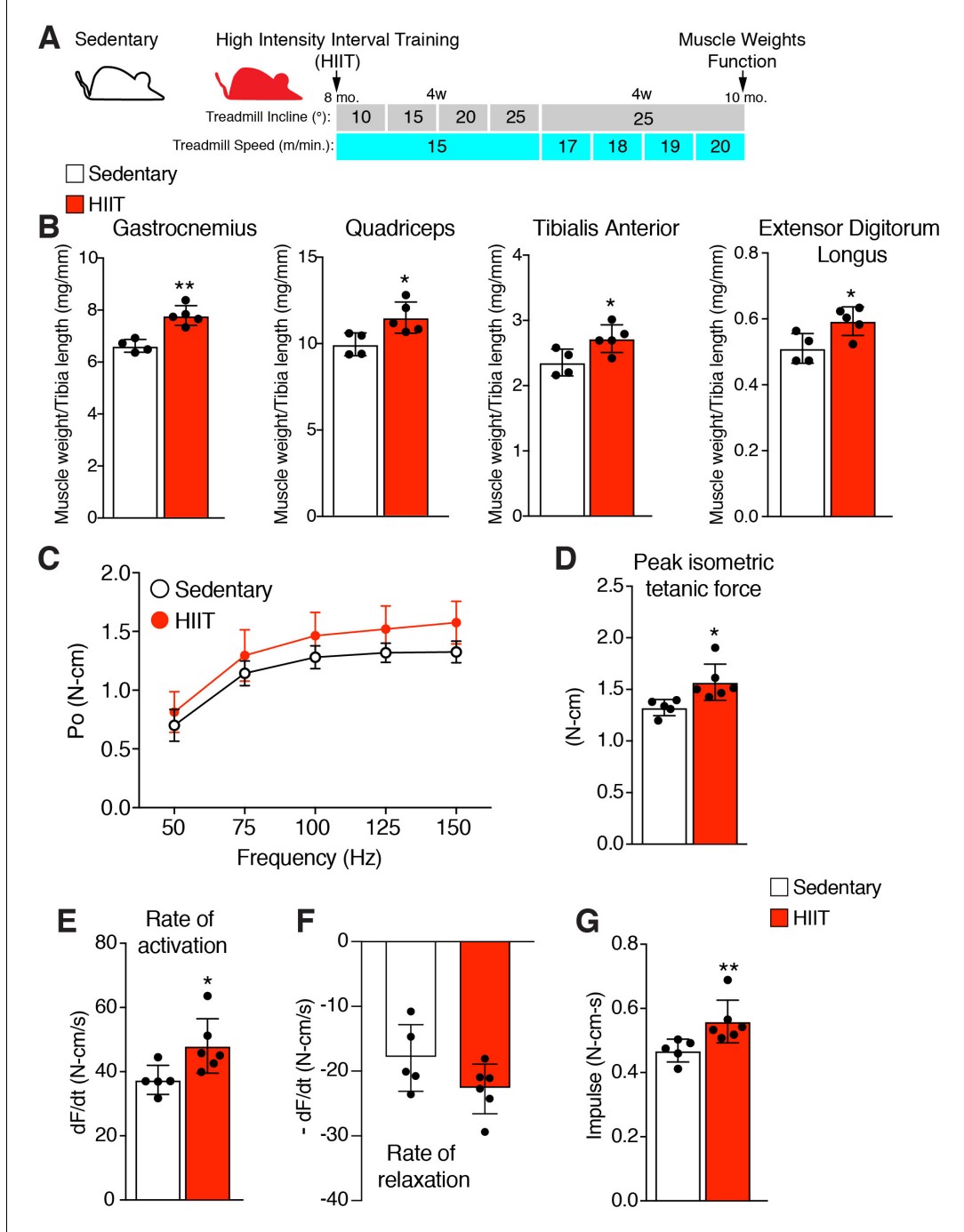

**Figure 1.** High intensity interval training (HIIT) in mice increases muscle size and strength. (**A**) Schematic showing the HIIT protocol using a treadmill. During the initial 4 weeks (w) treadmill incline was increased gradually each week while speed remained constant. In the final 4 weeks, incline was set at 25° but speed increased gradually each week. See Materials and methods for further details. (**B**) Muscle weight to tibia length ratios for multiple muscles in sedentary and HIIT-trained (8 weeks) mice. (**C–G**) In vivo isometric tetanic force measurements of plantar flexor muscles demonstrates functional improvements in muscle strength after 8 weeks of HIIT. (**C**) Force-frequency curve, (**D**) peak force, (**E**) rate of force generation, (**F**) rate of contraction relaxation, and (**G**) impulse. Data are presented as mean ± SD. Statistical analyses: (**B**), (**D**), (**E**) unpaired two-tailed Student's t-test, (**G**) Mann-Whitney test. *$p < 0.05$, **$p < 0.01$.

DOI: https://doi.org/10.7554/eLife.44876.002

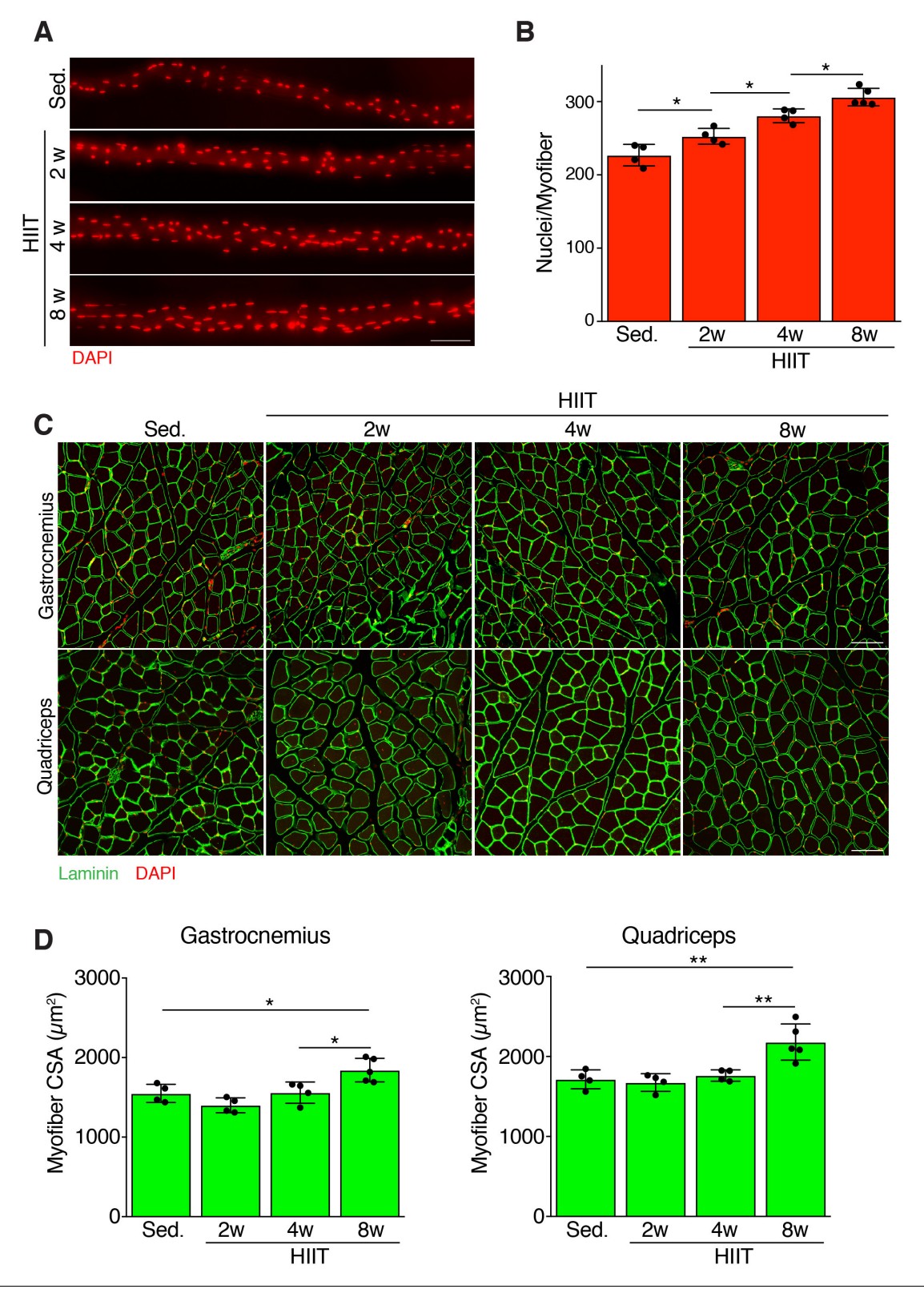

**Figure 2.** Persistent myonuclear accretion throughout HIIT while myofiber size increases between 4 and 8 weeks. (**A**) Representative single EDL myofibers from sedentary mice or mice subjected to HIIT for various durations. Myofibers were stained with DAPI to identify nuclei. (**B**) Total nuclei number per myofiber was quantified from the myofibers in (**A**) showing that myonuclear accretion occurs throughout the protocol. (**C**) Representative images of muscle sections from sedentary and HIIT-trained mice. Sections were immunostained with laminin antibodies and DAPI. (**D**) Quantification of

*Figure 2 continued*

myofiber cross-sectional area (CSA) in the gastrocnemius and quadriceps from the various groups of mice showing an increase of CSA 8 weeks after initiation of HIIT. Data are presented as mean ± SD. Statistical analyses: (B), (D) 1-way ANOVA with a Tukey correction for multiple comparisons. $*p<0.05$, $**p<0.01$. Scale bars: 100 μm.

DOI: https://doi.org/10.7554/eLife.44876.003

The following figure supplement is available for figure 2:

**Figure supplement 1.** An independent method to assess myonuclear number confirms a progressive increase during HIIT.

DOI: https://doi.org/10.7554/eLife.44876.004

## Loss of MP fusogenicity at the initiation of HIIT results in exercise intolerance and a failure to hypertrophy

Having established a protocol that leads to myonuclear accretion and myofiber hypertrophy, we next interrogated the role of MP fusogenicity during this physiological exercise regimen. Myomaker (*Mymk*) is an essential factor for fusion of MPs, and we have shown that it is required for growth associated with synergist ablation (*Goh and Millay, 2017*). Here we also used *Mymk*<sup>loxP/loxP</sup>; *Pax7*<sup>CreER</sup> mice to ablate fusion of MPs in an inducible manner prior to HIIT. *Mymk*<sup>loxP/loxP</sup>; *Pax7*<sup>CreER</sup>, and control *Mymk*<sup>loxP/loxP</sup> mice, were treated with tamoxifen (Tam.) at 8 months of age, and both groups of mice were subjected to HIIT for 8 weeks (*Figure 3A*). We refer to *Mymk*<sup>loxP/loxP</sup>; *Pax7*<sup>CreER</sup> in this set of experiments as Myomaker<sup>scKO 8w</sup> mice because MP fusion was ablated throughout the entire 8 weeks of HIIT. 4 weeks into HIIT it became apparent that Myomaker<sup>scKO 8w</sup> mice were not able to perform and properly adapt to the exercise stimulus. Indeed, quantification of the number of times a mouse landed on the shock grid at the bottom of the treadmill revealed a significant increase in occurrences by Myomaker<sup>scKO 8w</sup> mice, which progressively worsened throughout HIIT (*Figure 3B*). We also assessed myonuclear accretion and muscle morphometrics and function in Myomaker<sup>scKO 8w</sup> mice. Single EDL myofibers revealed similar numbers of nuclei between Myomaker<sup>scKO 8w</sup> sedentary and HIIT mice, indicating that myonuclear accretion was effectively blocked (*Figure 3—figure supplement 1A and B*). Analysis of muscle weights indicated no increase in size of multiple muscle groups between Myomaker<sup>scKO 8w</sup> sedentary and HIIT mice (*Figure 3—figure supplement 1C*). Moreover, histological examination of the gastrocnemius and quadriceps revealed that Myomaker<sup>scKO 8w</sup> mice did not undergo hypertrophy after 8 weeks of HIIT (*Figure 3C*). Quantification of myofiber CSA confirmed a lack of HIIT-induced hypertrophy in Myomaker<sup>scKO 8w</sup> mice (*Figure 3D*). Of note, we did observe a reduction of CSA in the quadriceps, but not gastrocnemius, of Myomaker<sup>scKO 8w</sup> sedentary mice suggesting a potential role for MP fusion during homeostasis. Finally, we also performed in vivo plantar flexion functional analysis and Myomaker<sup>scKO 8w</sup> HIIT mice did not display an increase in strength compared to Myomaker<sup>scKO 8w</sup> sedentary mice (*Figure 3—figure supplement 1D*). Taken together, loss of MP fusogenicity at the onset of HIIT results in exercise intolerance and a failure of the muscle to properly adapt.

## Altered response to HIIT depending on timing of ablation of MP fusogenicity

An inability for Myomaker<sup>scKO 8w</sup> mice to perform HIIT makes it difficult to determine the role of myonuclear accretion for the development of hypertrophy. Since initiation of an exercise regimen can lead to muscle damage that must be repaired to execute future bouts of exercise, and that Myomaker<sup>scKO 8w</sup> mice displayed exercise intolerance 3 weeks into HIIT, we suspected that HIIT may elicit muscle damage that requires MP fusion. Although exercise-induced muscle damage is not well-characterized, especially in comparison to muscle damage as a result of genetic diseases such as muscular dystrophy, release of creatine kinase is a standard assay to measure damage-associated loss of cell membrane integrity. We therefore assessed if HIIT elicited membrane damage that results in release of creatine kinase into the serum. Levels of creatine kinase in the serum of Myomaker<sup>scKO 8w</sup> mice were increased at 2 weeks and 4 weeks of HIIT compared to control mice (*Figure 3—figure supplement 2A*). Another defining aspect of exercise is the upregulation of genes in response to mechanical stress due to increased workload and thus we also evaluated expression of mechanical stress genes, including *Ankrd1*, *Ankrd2*, and *Csrp3* (*Mohamed et al., 2010*; *Barash et al., 2004*; *Chaillou et al., 2015*). Each of these contractile stress genes were upregulated

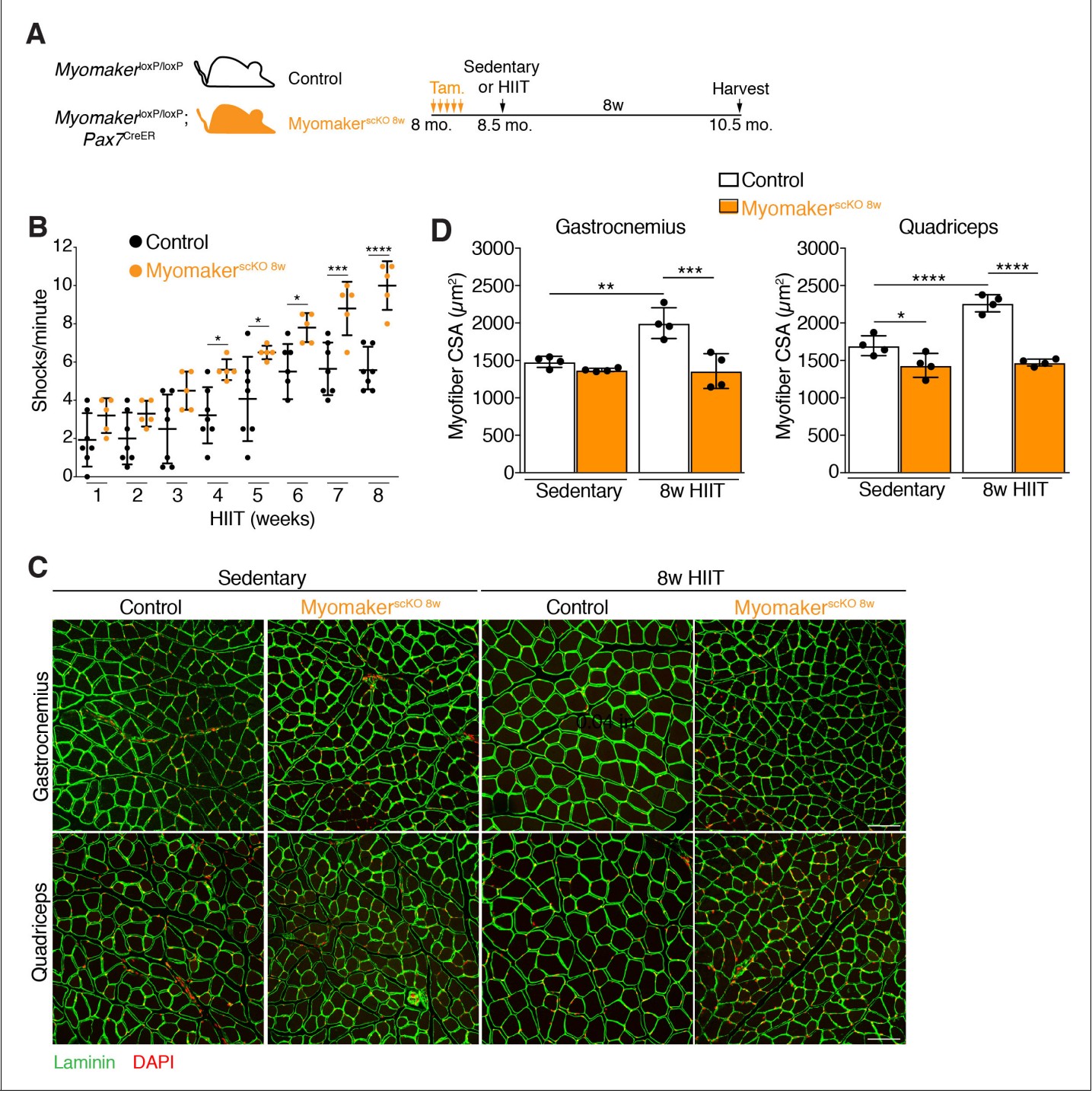

**Figure 3.** Genetic deletion of Myomaker in muscle progenitors at the onset of HIIT results in exercise intolerance and an absence of hypertrophy. (A) Illustration of experimental strategy to render muscle progenitors fusion-incompetent prior to HIIT. (B) Quantification of the number of times control and Myomaker^scKO 8w mice visited the shock grid at the bottom of the treadmill to assess the ability to accomplish the exercise protocol. (C) Representative sections from the gastrocnemius and quadriceps of control and Myomaker^scKO 8w mice after HIIT. Sections were immunostained with laminin antibodies and DAPI. (D) Quantification of myofiber CSA from the samples in (C). Data are presented as mean ± SD. Statistical analyses: (B), (D) 2-way ANOVA with a Bonferroni correction for multiple comparisons. *p<0.05, **p<0.01, ***p<0.001, ****p<0.0001. Scale bars: 100 μm.

DOI: https://doi.org/10.7554/eLife.44876.005

The following figure supplements are available for figure 3:

**Figure supplement 1.** Loss of Myomaker at the onset of HIIT effectively blocks myonuclear accretion and impairs subsequent functional gains.

*Figure 3 continued on next page*

*Figure 3 continued*

DOI: https://doi.org/10.7554/eLife.44876.006

**Figure supplement 2.** Muscle damage associated with HIIT fails to be resolved in Myomaker[scKO 8w] muscle.

DOI: https://doi.org/10.7554/eLife.44876.007

during HIIT in Myomaker[scKO 8w] mice (*Figure 3—figure supplement 2B, C and D*). These data indicate that the initial stages of HIIT led to some level of mechanical stress of the muscle, which was subsequently attenuated as part of the normal adaptive response in control mice. Thus, the inability to resolve the initial mechanical stress could account for the exercise intolerance observed in Myomaker[scKO 8w] mice. It should be noted that damage in Myomaker[scKO 8w] HIIT mice was not major enough to elicit functional deficits (*Figure 3—figure supplement 1C*). Nonetheless, MP fusion is required to adapt to mechanical damage during the early portion of HIIT, and a failure to repair this damage negatively impacts future bouts of exercise.

To avoid the requirement of MP fusion in the early stages of HIIT, which confounds analysis of hypertrophy at later stages, we treated *Mymk*[loxP/loxP]; *Pax7*[CreER] with tamoxifen 4 weeks into the HIIT protocol (*Figure 4A*). We refer to these mice as Myomaker[scKO 4w] mice because they are fusion-incompetent for the final 4 weeks of HIIT. We first tested if Myomaker[scKO 4w] mice were able to tolerate HIIT and found no difference in treadmill shocks per minute compared to control mice (*Figure 4B*). Since Myomaker[scKO 8w] mice exhibited exercise intolerance associated with elevated membrane and mechanical damage, we assessed whether these mice exhibit maladaptive remodeling of muscle, and then compared that response to Myomaker[scKO 4w] mice. We stained histological sections for Masson's Trichrome as an indicator of fibrosis. No evidence of fibrosis was observed in control or Myomaker[scKO] sedentary mice (*Figure 4C and D*). Control HIIT mice also displayed normal muscle architecture with the absence of fibrosis, but Myomaker[scKO 8w] mice exhibited interstitial fibrosis (*Figure 4C and D*) indicating that loss of MP fusion early in the HIIT protocol results in maladaptive remodeling. In contrast, no fibrosis was detected in Myomaker[scKO 4w] mice after HIIT showing that loss of MP fusion at this stage does not impact the ability of mice to exercise or result in maladaptive remodeling (*Figure 4C and D*). Moreover, these results reveal a direct link between loss of MP fusion in the early stages of HIIT and maladaptive remodeling.

Given that Myomaker[scKO 4w] mice exercised normally, and that HIIT-induced hypertrophy in control mice developed between 4 and 8 weeks of the protocol, we sought to determine if muscle growth was affected after ablation of MP fusion at late stages of HIIT (*Figure 5A*). Analysis of myofiber size in the gastrocnemius and quadriceps of control mice subjected to HIIT displayed the expected 15–20% increase, however Myomaker[scKO 4w] muscle failed to hypertrophy (*Figure 5B and C*). To confirm that MP fusion at 4 weeks of HIIT indeed resulted in reduced myonuclear accretion we assessed myonuclear numbers in isolated Myomaker[scKO 4w] EDL myofibers (*Figure 5D*). Compared to control mice that performed HIIT we observed a significant reduction of myonuclei in Myomaker[scKO 4w] myofibers, which reached the level of myonuclei in control mice at 4 weeks of HIIT (*Figure 5E*). This indicates that further myonuclear accretion was blocked soon after the 4 week time point in Myomaker[scKO 4w] muscle. We also assessed myofiber volume in EDL myofibers and observed larger myofibers in control HIIT compared to sedentary mice but did not detect any volume increase in Myomaker[scKO 4w] myofibers (*Figure 5F*). Taken together, HIIT-induced muscle growth requires MP fusion throughout the exercise protocol. While this indicates a tight relationship between myonuclear numbers and adaptive growth, it also raises the intriguing possibility that the mechanistic requirement for fusion at the later stages of HIIT is independent, and potentially distinct, from the need for accrual in the earlier stages that involve injury and repair.

One interpretation for the inability for Myomaker[scKO 4w] muscle to grow is that loss of MP fusogenicity has an acute effect on the growth potential of myofibers. To address this concern, we employed a pharmacologic approach to drive hypertrophy through manipulation of myostatin signaling, which is a negative regulator of muscle mass (*Lee, 2004*). We first treated *Mymk*[loxP/loxP] or *Mymk*[loxP/loxP]; *Pax7*[CreER] mice with tamoxifen at 3 months of age, to generate control and Myomaker[scKO] animals, respectively. Three days after the final tamoxifen dose we administered PBS or a soluble form of one of the myostatin receptors (activin type IIB) where the extracellular domain was fused to a Fc domain (ACVR2B-Fc) (*Lee et al., 2012*; *Lee et al., 2005*). Mice were treated weekly over 4 weeks and then muscle was harvested for assessment of hypertrophy (*Figure 5—figure supplement*

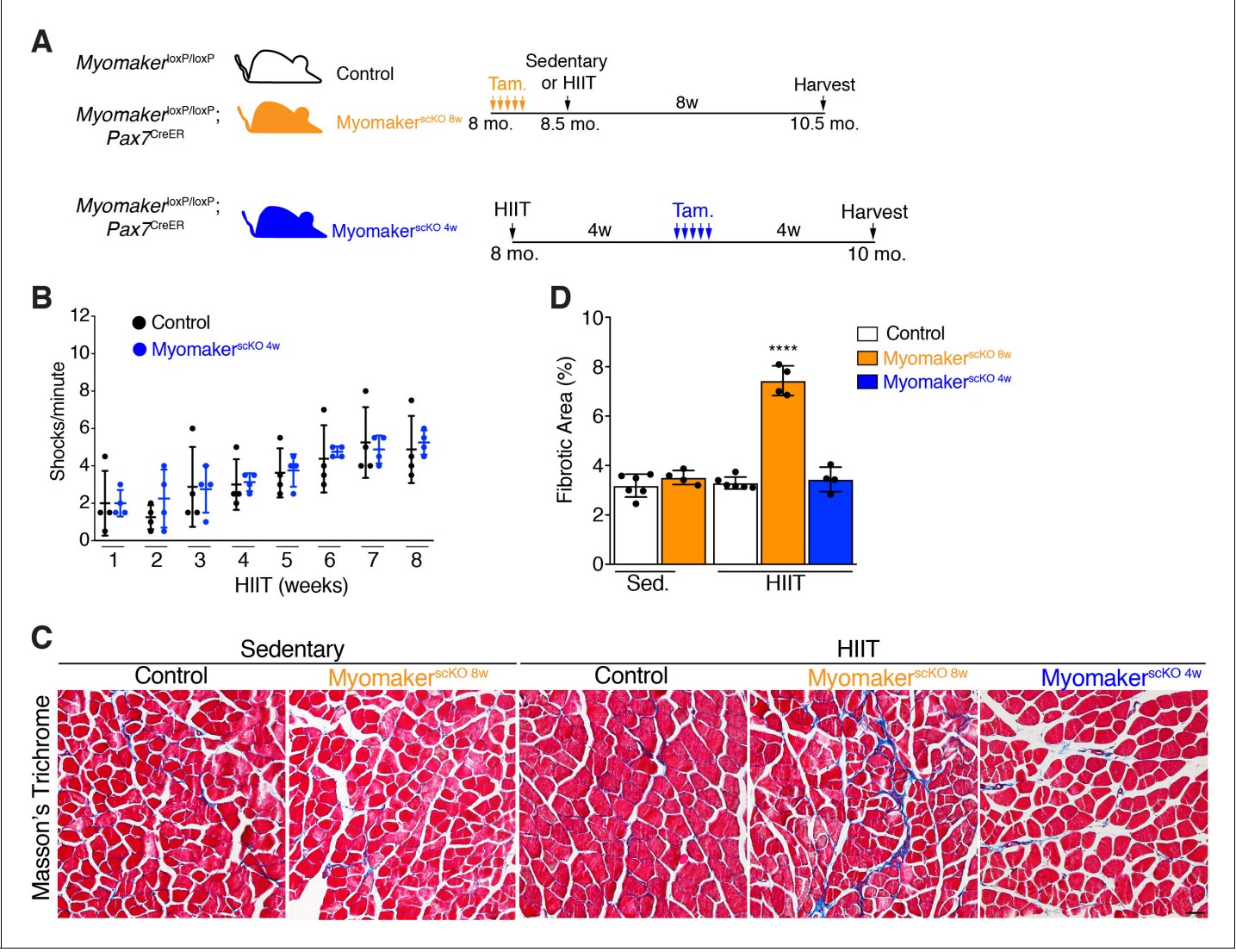

**Figure 4.** Loss of muscle progenitor fusogenic ability 4 weeks into HIIT does not affect exercise tolerance or result in maladaptive remodeling. (A) Schematic showing two different timelines of Myomaker ablation during HIIT. (B) Assessment of number of shocks per minute in control and Myomaker[scKO 4w] mice, which reveals that loss of Myomaker 4 weeks into HIIT had no impact on exercise tolerance. (C) Masson's trichrome staining shows that ablation of Myomaker at the beginning of HIIT (Myomaker[scKO 8w]) results in fibrosis indicating maladaptive remodeling. Evidence of fibrosis was not detected when Myomaker was deleted 4 weeks into HIIT (Myomaker[scKO 4w]). (D) Quantification of fibrotic area from the samples shown in (C). Data are presented as mean ± SD. Statistical analysis: (D) 1-way ANOVA with a Tukey correction for multiple comparisons. ****$p<0.0001$ compared to all other groups. Scale bar: 200 µm.

DOI: https://doi.org/10.7554/eLife.44876.008

*1A*). Histological analysis revealed an increase in myofiber size in both control and Myomaker[scKO] mice that received ACVR2B-Fc (*Figure 5—figure supplement 1B*). Assessment of muscle weights normalized to tibia length in ACVR2B-Fc-treated animals, as a percentage of their respective PBS-treated controls, also showed that Myomaker[scKO] muscle possesses the ability to grow in response to pharmacologic stimuli (*Figure 5—figure supplement 1C*). Muscle growth driven by myostatin inhibition in the absence of MP fusion ability, further indicates that the reason underlying a lack of hypertrophy in HIIT-trained Myomaker[scKO 4w] muscle is due to deficient myonuclear accretion in the final stages of exercise.

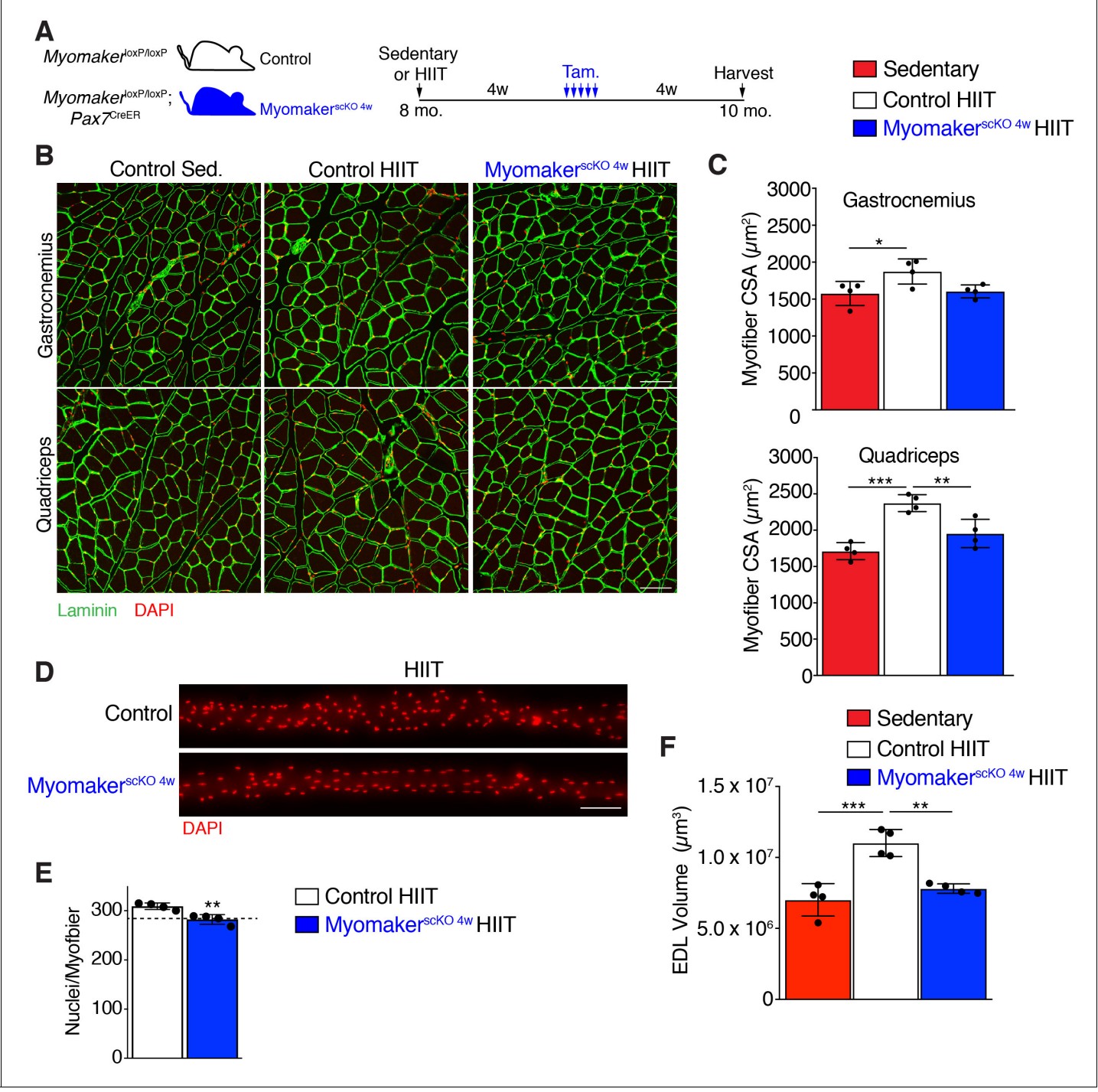

**Figure 5.** Myonuclei added over the final 4 weeks of HIIT are required for HIIT-induced hypertrophy. (**A**) Schematic showing how Myomaker[scKO 4w] mice were generated. (**B**) Representative sections of gastrocnemius and quadriceps from control sedentary (Sed.), control HIIT, or Myomaker[scKO 4w] HIIT mice. Sections were immunostained with laminin antibodies and DAPI. (**C**) Quantification of myofiber CSA from the samples in (**B**) showing that Myomaker[scKO 4w] myofibers fail to hypertrophy. (**D**) Individual myofibers from EDL demonstrating that Myomaker[scKO 4w] HIIT mice do not recruit as many myonuclei as control HIIT mice. (**E**) Quantification of myonuclei per myofiber from the samples in (**D**). The dotted line indicates the average myonuclear number of nuclei in WT myofibers after 4 weeks of HIIT (from *Figure 2A*) demonstrating that Myomaker[scKO 4w] HIIT mice fail to add new myonuclei after this time. (**F**) Assessment of myofiber volume in isolated EDL myofibers confirms WT HIIT-trained muscle undergoes hypertrophy, but this does not occur in Myomaker[scKO 4w] mice. Data are presented as mean ± SD. Statistical analyses: (**C**), (**F**) 1-way ANOVA with a Tukey correction for multiple comparisons, (**E**) unpaired two-tailed Student's t-test. *$p < 0.05$, **$p < 0.01$, ***$p < 0.001$. Scale bars: 100 μm.

DOI: https://doi.org/10.7554/eLife.44876.009

*Figure 5 continued on next page*

*Figure 5 continued*

The following figure supplement is available for figure 5:

**Figure supplement 1.** Myomaker[scKO] myofibers possess the ability to hypertrophy in response to a pharmacologic stimulus.
DOI: https://doi.org/10.7554/eLife.44876.010

## Discussion

In this study we evaluated the role of MP fusogenicity and associated myonuclear accretion during physiological exercise. Since the most often used exercise stimulus for mice, voluntary wheel running, does not lead to consistent hypertrophy we developed a HIIT protocol by exposing mice to a progressive increase in the amount of work performed through adjusting either the treadmill incline or speed over 8 weeks. We observed an increase in myonuclear numbers at each stage of HIIT indicating that MP fusion occurs throughout the exercise regimen. HIIT resulted in a 16% increase in myofiber CSA in the gastrocnemius and a 21% increase in the quadriceps, which developed between 4 and 8 weeks of the protocol. Ablation of MP fusion either at the onset of, or 4 weeks into, HIIT produces divergent observations with important ramifications. Absence of MP-derived Myomaker at the beginning of HIIT resulted in an inability to withstand the exercise stimulus, development of maladaptive remodeling, and lack of adaptive hypertrophy, whereas deletion of Myomaker 4 weeks into HIIT mainly impacted hypertrophy.

The requirement of MuSCs for myofiber hypertrophy has been debated for decades (*Snijders et al., 2015*; *Murach et al., 2018a*; *Cornelison, 2018*; *O'Connor et al., 2007*). Genetic perturbations of MuSCs and their MP descendants offer the most direct assessment of their role during muscle growth, but results have not led to a consensus paradigm. Ablation of MuSCs with a DTA allele in 4 month-old mice, which would inhibit the potential to generate fusogenic MPs, was initially reported to not impact the ability for myofibers to hypertrophy after synergist ablation, leading to the idea that MuSCs were dispensable (*McCarthy et al., 2011*). This experiment was repeated by an independent group that antithetically found MuSC-ablated muscle was unable to hypertrophy in 3–4 month-old-mice (*Egner et al., 2016*). We also performed synergist ablation on fusion-incompetent mice (through deletion Myomaker in MPs) at 3–4 months of age and found an absolute requirement of MP fusogenicity for growth (*Goh and Millay, 2017*). Moreover, another study confirmed the requirement of MuSCs for overload-induced hypertrophy in 3 month old mice through genetic deletion of serum response factor (Srf), which is required for fusion (*Randrianarison-Huetz et al., 2018*). Current ideas to explain the discrepancies are that young mice (younger than 4 months of age) are reliant on MP fusogenicity for growth but older myofibers can hypertrophy in the absence of new myonuclei (*Murach et al., 2017b*). Recent evidence claims an age-dependence on MuSCs as ablation of MuSCs at 4 weeks of age reduces normal developmental hypertrophy (*Bachman et al., 2018*), while loss of MuSCs in the adult has minimal effects in sedentary mice (*Fry et al., 2015*; *Keefe et al., 2015*). We show here that ablation of MP fusion at the onset of HIIT in 8-month-old mice results in exercise intolerance and fibrosis. Since synergist ablation is a stronger stimulus compared to HIIT, we think it is unlikely that Myomaker[scKO] mice at 8 months of age would be able to properly adapt to synergist ablation. Overall, our data indicates that age dependence may not sufficiently explain the discrepancy about the role of MuSCs for synergist ablation-induced hypertrophy in mice that are 3–4 months of age.

Myonuclear accretion during HIIT is continuous where approximately 25 myonuclei per myofiber are added within the first 2 weeks, another 25–30 myonuclei between 2 and 4 weeks, and then 25 myonuclei over the final 4 weeks. While myonuclear number increases throughout HIIT, growth of myofibers occurred between 4 and 8 weeks indicating that myonuclear accretion precedes detectable growth. This is consistent with the response of muscle to synergist ablation where myofibers acquire new nuclei before an increase in size (*Bruusgaard et al., 2010*). Our data not only demonstrate the persistent nature of myonuclear accrual during exercise, but we also shed light on whether myofibers require new nuclei throughout the protocol to grow. Ablation of Myomaker in MPs 4 weeks after HIIT resulted in a failure to hypertrophy even though this only corresponds to a lack of 25 myonuclei. These data show that the addition of 50–55 myonuclei added during the initial 4 weeks of HIIT are not sufficient to drive adaptive hypertrophy. Thus, a key finding from our work is that HIIT-induced adaptive growth requires continued myonuclear accrual throughout the protocol.

Exercise-induced adaptations involve the response to myofiber damage, increased contractile output, and growth of the myofiber, but it has been difficult to ascertain if myonuclear accrual is required for each of these adaptations. Surprisingly, the HIIT protocol described here may have uncoupled these adaptive processes since there are multiple lines of evidence suggesting distinct phases (mitigation of injury and growth) during HIIT. First, we show that muscle damage (assessed by levels of serum creatine kinase and contractile stress genes) is a normal response to HIIT and these damage markers progressively diminish over the duration of the protocol. If fusion is abrogated at the onset of HIIT (Myomaker$^{scKO\ 8w}$), mice are unable to repair HIIT-induced muscle damage, which causes exercise intolerance. However, when mice are allowed to accumulate new nuclei into the myofiber (approximately 50–55) during the first four weeks but further MP fusion is blocked (Myomaker$^{scKO\ 4w}$), the mice are able to exercise normally indicating that the damage phase of exercise has been mitigated. These data are generally consistent with reports in humans that resistance training causes early increases in protein synthesis to support tissue repair but does not correlate with hypertrophy, which occurs only after the repair of damage (*Damas et al., 2016*; *Mitchell et al., 2014*; *Moore et al., 2005*).

What could explain the observation that Myomaker$^{scKO\ 4w}$ mice are able to exercise normally during the final four weeks of HIIT? One possibility is that the nature of the stimulus changed from a gradual increase in treadmill incline in the first four weeks to an increase in speed (with constant incline) over the next four weeks. One argument against this possibility is that Myomaker$^{scKO\ 8w}$ mice exhibit progressive failure to exercise even after the protocol changed. A limitation of our study is that we did not test if Myomaker$^{scKO\ 4w}$ mice eventually become exercise intolerant by extending the exercise protocol or performing an acute treadmill run-to-exhaustion test. Another possibility that could explain the ability for Myomaker$^{scKO\ 4w}$ mice to exercise is that nuclei added during the first four weeks of HIIT still support the reparative response in the final four weeks of HIIT, which allows them to tolerate the exercise.

Since Myomaker$^{scKO\ 4w}$ mice exercise normally during the final four weeks of HIIT and have accumulated 50–55 new nuclei/fiber, why do they not undergo muscle hypertrophy? This question is currently difficult to answer but dogma would argue that Myomaker$^{scKO\ 4w}$ muscle has not accumulated enough myonuclei to grow. Perhaps there is a tight correlation between absolute myonuclear numbers and growth potential, with minimal tolerance for reduced numbers. A more provocative explanation is that the added myonuclei may be assigned specialized roles depending on the stage of adaptation. In this scenario, the 25 new myonuclei added in the final 4 weeks of HIIT are specialized for growth whereas the 50–55 myonuclei added in the first 4 weeks of HIIT are mainly utilized for repair of tissue injury. Adding support to such an idea of specialized roles of accrued myonuclei is that our data also reveal a potential lack of redundancy within the population of the newly accrued nuclei, given the apparent inability of the 50–55 nuclei added early in HIIT to compensate for the lack of 25 nuclei late in HIIT to drive growth. The two ideas discussed here, that there is a threshold of myonuclear number that must be met for growth and the potential for myonuclear functional heterogeneity, are not necessarily inconsistent with each other. More specifically, it is possible that there is a tight correlation of myonuclear number for growth because new nuclei specialized for growth are required, thereby increasing overall myonuclear numbers. Precise mechanisms by which myofibers utilize the new nuclei during various stages of adaptation warrants further investigation.

The importance of myonuclear accretion has classically been viewed through the lens of the myonuclear domain theory, which postulates that a given myonucleus controls a certain volume of cytoplasm (*Murach et al., 2018b*), and that for muscle to grow it needs more nuclei to support the larger cellular volume. We propose that the myonuclear domain theory may be too simplistic to explain the complexity of muscle adaptability. A major issue with the myonuclear domain theory is that it links myonuclear numbers solely with growth. In doing so, it assumes all myonuclei, freshly accrued or added during development, are homogenous in terms of both magnitude and profile of transcriptional activity. However, myonuclei possess the ability to alter transcription in the absence of MuSCs suggesting a dynamic, heterogenous system (*Kirby et al., 2016b*). Our data reveals that myonuclear accrual extends beyond just the maintenance of the myonuclear domain. Indeed, MP fusion and myonuclear accrual are essential for normal exercise-related adaptive responses that encompass both repair and growth, which has implications for maintaining muscle mass during disease and aging.

# Materials and methods

**Key resources table**

| Reagent type (species) or resource | Designation | Source or reference | Identifiers | Additional information |
|---|---|---|---|---|
| Genetic reagent (M. musculus) | $Mymk^{loxP/loxP}$ | *Millay et al., 2014*; *Goh and Millay, 2017* | | Dr. Douglas Millay (Cincinnati Children's Hospital Medical Center) |
| Genetic reagent (M. musculus) | $Pax7^{CreER}$ | *Lepper et al., 2009* | | Dr. Chen-Ming Fan (Carnegie Science) |
| Peptide, recombinant protein | ACVR2B-Fc | *Lee et al., 2012* | | Dr. Se-Jin Lee (The Jackson Laboratory) |

## Animals

*Mymk* mouse strains used in this study have been described previously (*Goh and Millay, 2017*; *Millay et al., 2014*). Briefly, fusion-incompetent mice (Myomaker$^{scKO}$) were generated by breeding $Mymk^{loxP/loxP}$ mice with satellite cell-specific ($Pax7^{CreERT2}$) Cre recombinase mice (*Lepper et al., 2009*), while $Mymk^{loxP/loxP}$ mice served as controls. All experiments were performed on male mice. All animal procedures were approved by Cincinnati Children's Hospital Medical Center's Institutional Animal Care and Use Committee (IACUC2017-0053).

## Tamoxifen treatment

Tamoxifen (TAM; Sigma-Aldrich) was mixed in corn oil with 10% ethanol at 25 mg/ml. Prior to initiation of 8 weeks of HIIT, 8 month old mice were administered five daily doses of TAM at 0.075 mg/g/day through intraperitoneal injections (Myomaker$^{scKO\ 8w}$). In a second group of 8 month old mice, TAM treatment was delayed, and the five daily doses were administered after the 4$^{th}$ week of HIIT (Myomaker$^{scKO\ 4w}$). After initial TAM dosing, all mice received a weekly TAM injection for the remaining duration of the HIIT protocol. For the ACVR2B-Fc study, 3 month old mice were subjected to five daily doses of TAM 3 days prior to administration of ACVR2B-Fc, and maintained on a TAM regimen (injections every three days) for 4 weeks.

## High-intensity interval training (HIIT)

We modified a recently reported high-intensity interval training protocol (HIIT) for aged mice (*Seldeen et al., 2018*). Our 8 week HIIT protocol was designed to increase the amount of work performed each week. For the first 4 weeks of HIIT, there was a weekly 5° increase in the angle of inclination from 10° at week 1 up to 25° at week 4. Treadmill speed was maintained at 15 m/min for this duration. From weeks 5–8, treadmill speed was increased by 1 m/min weekly to a maximum speed of 20 m/min at week 8, while the angle of inclination was maintained at 25°. Using the equation developed previously (Work (J) = body mass (kg) x gravity (9.81 m/sec$^2$) x vertical speed (m/sec x angle) x time (sec)) (*Nie et al., 2016*) to determine the amount of work performed (J), we established that our HIIT protocol results in a weekly increase of 5–15% in work.

Prior to HIIT, mice were acclimated on the treadmill three times a week at a speed of 8 m/min for 15 min with no incline. After a week of acclimation, mice were subjected to 3 sessions of HIIT per week for 8 weeks. Each session consisted of a 5 min warm-up at 8 m/min, eight exercise intervals at the prescribed speed and angle of inclination for 3–5 min, and a 1 min rest interval at 8 m/min between each exercise interval. As a negative reinforcement, mice that were unable to stay on the treadmill were shocked by an electric stimulus emanating from a shock grid located at the back of the treadmill. The electric stimulus was set at a current of 1 mA and a repetition rate of 2 Hz. At the last HIIT session of each week, exercise tolerance in each mouse was assessed by counting the number of times the mouse landed on the shock grid. Each mouse was visually observed for at least 5 min for this purpose. To ensure mice could meet the demands of the protocol, we established an exclusion-criteria based on the inability of the mouse to recover following electric shock exposure.

Mice were given 5 s to resume running after landing on the shock grid; those that remained on the shock grid for more than 5 s were manually assisted back on the treadmill. If such assistance occurs more than twice during an exercise interval, the mouse is excluded from treadmill running for the duration of that interval. An exclusion from two exercise intervals in the same HIIT session would result in the mouse being excluded from the remainder of the session. Finally, an exclusion from more than 2 HIIT sessions resulted in expulsion from the study and no data from these mice were used for analysis. Based on this exclusion criteria, we had to remove 1 Control mouse, 3 Myomaker[scKO 8w], and 0 Myomaker[scKO 4w] mice. All HIIT sessions were performed on an Exer 3/6 Treadmill from Columbus Instruments.

## Inhibition of myostatin signaling

Inhibition of myostatin/activin A signaling, a potent negative regulator of muscle mass, was achieved by a pharmacological approach previously described by others (*Lee et al., 2012*). Briefly, robust muscle hypertrophy was induced by utilizing a soluble form of the activin type IIB receptor where the extracellular ligand binding domain has been fused to an Fc domain (ACVR2B-Fc), effectively rendering it as a decoy receptor for myostatin. 3 month old mice were given four weekly doses of PBS or ACVR2B-Fc at 10 mg/kg through intraperitoneal injections 3 days after the last dose of TAM was administered.

## In vivo muscle force measurements

Following 8 weeks of HIIT, the strength of plantar flexor muscles was evaluated by in vivo isometric tetanic force measurements. Mice were placed under anesthesia with 2% isoflurane inhalation, then the right knee was fixed on a clamp, and the right foot secured at a 90° angle on the pedal of a dual-mode servometer (300C-LR: Aurora Scientific, Aurora, ON, Canada). To stimulate the tibia nerve, needle electrodes connected to an electric stimulator (701C: Aurora Scientific, Aurora, ON, Canada) were inserted in the posterior knee. Isometric twitch force (Pt) was generated by a 0.2 ms pulse with 50 mA to find the optimal muscle length. Maximum isometric tetanic force (Po) of the plantar flexor muscles was subsequently evaluated by a 0.2 ms pulse for 350 ms at a frequency of 50 to 150 Hz. A two-minute rest was allowed between each tetanic contraction. All force measurement data was collected by Dynamic Muscle Control (DMC v5.5) and analyzed based on the tetanic force graph by Dynamic Muscle Analysis (DMA v5.3), which yielded peak tetanic force, rate of force activation and relaxation, and impulse achieved during a tetanic contraction.

## Muscle collection and preparation

Quadriceps, gastrocnemius, tibialis anterior (TA), and extensor digitorum longus muscles (EDL) were dissected, blotted dry, and weighed. Hindlimbs were dissolved overnight in lysis buffer containing 0.4 mg/ml proteinase K (Sigma Aldrich #03115879001) at 55°C. Tibia length was assessed by a digital caliper, and the average of three measurements was recorded. For histological sections, muscles were embedded in OCT compound (gastrocnemius and quadriceps) or 1% tragacanth/PBS (TA) and frozen in liquid nitrogen-cooled 2-methylbutane for cryosections, or embedded in warm paraffin wax (gastrocnemius) for paraffin sections that were stained with Masson's Trichrome. Sections were then cut at 10 µm. For gene expression analysis, quadriceps were flash-frozen in liquid nitrogen and stored at –80°C. For single fiber analysis, whole EDL muscles were incubated in 0.2% Type I collagenase (Sigma-Aldrich #C-0130) in DMEM (Hyclone) for 1 hr at 37°C, and then transferred to DMEM containing 10% horse serum and triturated to release individual fibers.

## Histological analyses and serum creatine kinase

Cryosections were fixed in 1% PFA/PBS, permeabilized with 0.2% Triton X-100/PBS, blocked with 1% BSA/PBS/1% heat-inactivated goat serum/0.025% Tween20/PBS, and incubated with primary antibody overnight at 4°C. Following 1 hr of incubation with a secondary Alexa Fluor antibody (1:200) (Invitrogen), slides were mounted with VectaShield containing DAPI (Vector Laboratories). For assessment of hypertrophy, samples were stained with anti-laminin (1:100; Sigma-Aldrich # L9393), and myofiber cross-sectional area (CSA) was visualized and quantified as described previously (*Goh and Millay, 2017*). To assay for myonuclear number, samples were stained with anti-dystrophin (1:100; Abcam #ab15277), and visualized at 40x on a Nikon A1R + LUNV on a Ti-E Inverted

Microscope. The total number of DAPI$^+$ nuclei within a dystrophin-stained myofiber was quantified. Myonuclear accretion was also assessed in single EDL myofibers. Isolated fibers were fixed in 4% PFA/PBS for 30 min at 4°C, washed three times in PBS, permeabilized with 0.2% Triton X-100/PBS and mounted with Vectashield containing DAPI. 10x optical sections were generated on a Nikon Ti-E SpectraX widefield microscope, and the total number of DAPI$^+$ nuclei along the entire length of a myofiber was quantified with a spot detection algorithm developed in Imaris (Bitplane). To quantify the size of a single fiber, length and radius measurements were performed in Imaris, and muscle volume was calculated. For assessment of fibrosis, Masson's Trichrome staining was performed on paraffin sections using standard protocols. Fibrotic area within a section was visualized at 20x with an Olympus BX51 inverted microscope, and quantified with a binary thresholding algorithm from Imaris (Bitplane). Blood was harvested from the inferior vena cava of anesthetized mice, then submitted to the clinical laboratory at Cincinnati Children's Hospital Medical Center for analysis of creatine kinase in the serum.

## RNA analysis

Total RNA was extracted from quadriceps muscle with Trizol (Invitrogen), and cDNA was synthesized using MultiScribe reverse transcriptase with random hexamer primers (Applied Biosystems). Expression of contractile stress genes was assessed using standard qPCR approaches with PowerUp SYBR Green Master Mix (Applied Biosystems). Analysis was performed on a 7900HT fast real-time PCR machine (Applied Bio-systems) with the following primers: *Ankrd1* (Forward, 5'-GGATGTGCCGAGG TTTCTGAA-3' and Reverse, 5'-GTCCGTTTATACTCATCGCAGAC-3'); *Ankrd2* (Forward, 5'-TGGACA TGCTAGTGCTAGAGG-3' and Reverse, 5'-CGCTTTTTCTGCTTGCGTTTT-3'); *Csrp3* (Forward, 5'-TC TACTGTAAGGTGTGCTATGGG-3' and Reverse, 5'-GCTTTGGGGATTGTTGGAACTG-3'). *GAPDH* primers (Forward, 5'-TGCGACTTCAA-CAGCAACTC-3' and Reverse, 5'-GCCTCTCTTGCTCAGTG TCC-3') were used as internal controls.

## Statistical analysis

All quantitative data sets that contained at least four samples were initially assessed with a Shapiro-Wilk normality test to determine distribution of values. Differences in values between two data sets were analyzed with an unpaired two-tailed Student's $t$-test (normal distribution) or Mann-Whitney rank sum test (non-normal distribution) using GraphPad Prism 7.0 software. We determined variance for each data set with a F test and all data sets showed equal variance, therefore tests to correct for unequal variance were not needed. For multiple data sets with one independent variable, a 1-way ANOVA with Tukey correction for multiple comparisons was performed. For data sets with two independent variables, a 2-way ANOVA with Bonferroni correction for multiple comparisons was performed. Statistical significance was set at a P value < 0.05. For assessments of size (myofiber CSA and single fiber muscle volume), total nuclei (cross-sections and single fibers), and fibrosis, data analyses were performed in a blinded fashion. All data sets are presented as means $\pm$ SD, and the degree of significance between data sets is depicted as follows: *p<0.05, **p<0.01, ***p<0.001, ****p<0.0001.

## Acknowledgements

We thank the Confocal Imaging Core at Cincinnati Children's Hospital for assistance with Imaris-based algorithms, and Vikram Prasad of the Molkentin Laboratory for helpful discussions and experimental advice. We thank Dr. Daniel Schnell (Cincinnati Children's Hospital) for advice on statistical analyses. This work was supported by grants to DPM from the National Institutes of Health (NIH) (R01AR068286, R01AG059605) and the Pew Charitable Trusts. SJL is supported by a grant from the NIH (R01AR060636). SS receives support from NIH grants (R01HL130356, R01HL105826, R01AR067279 and RO1/R56HL139680).

## Additional information

### Funding

| Funder | Grant reference number | Author |
|---|---|---|
| National Institutes of Health | R01AR068286 | Douglas P Millay |
| Pew Charitable Trusts | | Douglas P Millay |
| National Institutes of Health | R01AG059605 | Douglas P Millay |
| National Institutes of Health | R01AR060636 | Se-Jin Lee |
| National Institutes of Health | R01HL130356 | Sakthivel Sadayappan |
| National Institutes of Health | R01HL105826 | Sakthivel Sadayappan |
| National Institutes of Health | R01AR067279 | Sakthivel Sadayappan |
| National Institutes of Health | RO1/R56HL139680 | Sakthivel Sadayappan |

The funders had no role in study design, data collection and interpretation, or the decision to submit the work for publication.

### Author contributions

Qingnian Goh, Conceptualization, Data curation, Formal analysis, Investigation, Methodology, Writing—review and editing; Taejeong Song, Michael J Petrany, Alyssa AW Cramer, Data curation, Writing—review and editing; Chengyi Sun, Data curation, Methodology, Writing—review and editing; Sakthivel Sadayappan, Data curation, Formal analysis, Funding acquisition; Se-Jin Lee, Methodology, Writing—review and editing; Douglas P Millay, Conceptualization, Formal analysis, Funding acquisition, Investigation, Methodology, Writing—original draft, Project administration, Writing—review and editing

### Author ORCIDs

Alyssa AW Cramer (iD) http://orcid.org/0000-0003-2997-5066
Chengyi Sun (iD) http://orcid.org/0000-0001-8500-1878
Douglas P Millay (iD) http://orcid.org/0000-0001-5188-0720

### Ethics

Animal experimentation: This study was performed in strict accordance with the recommendations in the Guide for the Care and Use of Laboratory Animals of the National Institutes of Health. All of the animals were handled according to approved institutional animal care and use committee (IACUC) protocols of the Cincinnati Children's Hospital Medical Center.

### Decision letter and Author response

Decision letter https://doi.org/10.7554/eLife.44876.014
Author response https://doi.org/10.7554/eLife.44876.015

## Additional files

### Supplementary files

• Transparent reporting form
DOI: https://doi.org/10.7554/eLife.44876.011

### Data availability

All data generated in this study are included in the manuscript and supporting files.

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
