## [Decision Letter]

Thank you for submitting your article "Myonuclear accretion is a determinant of exercise-induced remodeling in skeletal muscle" for consideration by *eLife*. Your article has been reviewed by two peer reviewers, and the evaluation has been overseen by a Reviewing Editor and Sean Morrison as the Senior Editor. The following individuals involved in review of your submission have agreed to reveal their identity: Gabrielle Kardon (Reviewer #2).

The reviewers have discussed the reviews with one another and the Reviewing Editor has drafted this decision to help you prepare a revised submission.

Summary:

The manuscript from Goh et al. directly addresses a fundamental controversy on how muscle mass is built. More specifically if fusion of activated satellite cells (MP) are mandatory for hypertrophy. Reviewers felt that the approaches are unique and definitive. While there is excitement for the manuscript, there was a concern regarding the number of mice used in some of the experiments, During the discussion between the BRE and reviewers, it was felt that increasing the numbers of mice in these data sets would be desirable-if they are available. If not, more careful use of the statistics and more cautious conclusions.

Below are the main points to consider. These points are given more context within the reviewers individual comments.

1) The major weakness of the study is that the number of observations is low (down to 3), and some of the results therefore need to be interpreted with caution. Can more mice be included into specific experiments?

2) The authors make conclusions regarding the phases (first 4 weeks versus 2nd 4 weeks) of HIIT, but there may be an alternative interpretation. Could these be different responses to increasing incline versus increasing speed?

3) The authors discuss muscle damage a lot, but the molecular data they provide (Figure 3—figure supplement 2) is not really on molecules related to damage or repair/regeneration: heat-shock proteins, embryonal myosin or creatine kinase would be beneficial.

4) Clarification and caution in the Discussion.

*Reviewer #1:*

This is a very interesting paper that deal with some of the fundamental controversies on how muscle mass is built. More specifically, if fusion of activated satellite cells (MP) are mandatory for hypertrophy. There have been two major caveats in this debate, one is that most of the conclusions are based on supra-physiological overload by surgical ablation of synergists. The other is a current perception that only in animals less than 4 months of age muscle hypertrophy is dependent on MP fusion. The current paper resolves both these questions: animals of 8 months of age subjected to a physiological high intensity training protocol displayed significant hypertrophy, while when MP fusion was prevented, hypertrophy was also prevented. In my opinion these are the most interesting findings, but the paper also demonstrated that when MP fusion was prevented at the onset of exercise the animals were unable to fulfil the exercise task and muscle fibrosis was also induced. If the fusion was prevented only after the initial exercise phase these phenomena did not occur, but hypertrophy was still largely prevented.

A particular strength, is that the authors has dealt elegantly with the problem of the transgenic animals not performing well on the exercise task and that they displayed fibrosis.

The major weakness of the study is that the number of observations is low (down to 3), and some of the results therefore need to be interpreted with caution. The statistics should be improved or at least be described better. The authors might benefit from consulting a statistician.

Another weakness is the Discussion seems to create strawmen that the authors argue against, the Discussion should be rewritten.

Specific points:

Statistics

The authors use a mix of parametric and non-parametric testing, they need to make clear what the criteria for selecting each method was, and which p-values are derived from which method (for example in the legends).

Is it possible to determine from 3 observations if they are normally distributed or not?

Should non-parametric testing be accompanied with median and percentiles rather than mean an SD?

There seems not that much difference in variance between groups, why was Welch test used?

How about corrections for multiple comparisons?

Would the time courses be better tested by using nested ANOVA, two-way ANOVA or a linear mixed analysis?

Results

The authors discuss muscle damage a lot, but the molecular data they provide (Figure 3—figure supplement 2) is not really on molecules related to damage or repair/regeneration but rather to mechanical stress which might not always damage. If the authors still have extracts of muscles, they should be able to measure more relevant molecules such as heat-shock proteins or embryonal myosin (the latter can also be analysed on cryosections). It would also be useful if they could compare such markers between control and experimental groups, e.g. is the level higher in the mice that were tamoxifen injected at the onset than for other groups (controls and late injected.

Discussion

As mentioned, I find large parts of the Discussion problematic, for example:

"Exercise-induced adaptations involve the response to myofiber damage, increased contractile output, and growth of the myofiber, but it has been difficult to ascertain if myonuclear accrual is required for early exercise-related injury alone, adaptive growth alone, or both."

I have never seen anybody suggest that myonuclear accrual should not be related to repair of injury. Very unclear statement, rephrase.

"Surprisingly, the HIIT protocol described here uncoupled these adaptive processes."

I am not sure the data really show this. You would need to explain better or delete.

"We observed an increase in myonuclear numbers at each stage of HIIT indicating that MP fusion occurs and that accrual is not a consequence of a single event."

I am not sure I understand this. What single event? Do you mean onset of exercise? I would interpret the data to mean that there is a gradual increase in the number of nuclei with a somewhat lagging increase in CSA during exercise. Bruusgaard et al., 2010 showed by overload that the MP fusion seemed to precede the CSA growth, and thus hinting on a causal relationship. When the authors here state "It is interesting to consider that in WT mice myonuclear accretion occurs throughout HIIT but hypertrophy transpires at later stages." It sounds like a similar conclusion. Could the authors plot the data in a better way illustrating this point?

"This suggests that an increase in myonuclei does not necessarily indicate that hypertrophy is forthcoming, which could explain the inconsistencies of studies reporting a correlation between myonuclear number and myofiber size and other studies finding no correlation (Murach et al., 2018; Damas et al., 2018)."

I don't understand this argument. I have not seen anybody suggest that an increase in myonuclei is in itself is sufficient to induce hypertrophy. As I understand it the general idea has been that more myonuclei might be required for hypertrophy, but that some hypertrophic signal (for example load) is also required. For example, inactive nulei-rich fibres will also atrophy if denervated. I also don't understand how the findings of Murach et al., 2018 (is a review not a study) and Damas et al., 2018, could be explained in this way?

"Since our data indicate that evidence of myonuclear accretion cannot be used simply to predict myofiber size, we propose that addition of myonuclei may serve specialized roles depending on the stage of adaptation. For instance, myonuclei could be recruited to the myofiber during exercise initiation to allow adaptation to moderate damage whereas myonuclear accretion at the later stages of HIIT could facilitate adaptive hypertrophy. Collectively, these observations agree with reports in humans that resistance training-induced hypertrophy occurs after suppression of damage (Damas et al., 2016)".

My interpretation would be that muscle fibres without new nuclei don't increase sufficiently in size and/or strength to cope with the forced HIIT and some of the fibres are ruptured. Muscle damage in untrained people that start to exercise abruptly and hard is well known (rhabdomyolysis), and here electroshock is used. I don't think this translates to specialized roles for nuclei depending on stage of adaption or that early accretion is necessarily related to damage. Has damage been observed in the control mice here? The title of Damas et al., 2016, rephrased here is misleading. As I understand it, the authors of that paper did not really suppress damage, they just compared synthesis during early phases with lots of repair and later phases with less repair and more radial growth. If this part is to be kept in the Discussion the relevance to the present paper needs to be explained better.

*Reviewer #2:*

Muscle stem cells, known as satellite cells, are well known to be required for regeneration after acute muscle damage. However, the role of satellite cells during muscle hypertrophy has been controversial. Most studies using mouse to examine the contribution of satellite cells during hypertrophy have genetically ablated satellite cells, using Pax7^CreERT2^/+;RosaDTA/+ mice, in the context of synergistic ablation. However, synergistic ablation may not be physiologically relevant to exercise-induced hypertrophy and satellite cell ablation may have unknown consequences on the molecular and cellular characteristics of the myofiber (for instance, if the myofiber somehow can "sense" the presence of satellite cells and so the absence of satellite cells may change the myofiber characteristics). This manuscript from the Millay lab takes a new approach toward interrogating the role of satellite cells in muscle hypertrophy. First, they develop a new system to induce muscle hypertrophy in mice: 8 month old mice are subject to running 3x/week (at 15-16 meters/minute) on a treadmill in which the incline angle progressively increases (10o, 15o, 20 o, and 25 o) during the first 4 weeks and then the treadmill speed increases progressively (17, 18, 19, and 20 m/min) during the final 4 weeks. Second, in experimental mice they specifically prevent fusion of satellite cells to the myofibers using the tightly controlled Pax7^CreERT2^/+;*Mymk*^loxP/loxP^, in which Myomaker (required for fusion) is deleted inducibly in satellite cells. Using these two techniques. they gain unique insights into the role of satellite cells in hypertrophy. First, they find that their High Intensity Interval Training (HIIT) protocol does indeed lead to myofiber hypertrophy during 4-8 weeks of HIIT (as measured by an increase in myofiber cross-sectional area of 4 different leg muscles) with increased muscle strength and this is accompanied by an increase in myonuclei/myofiber (as measured in EDL and Quadriceps muscles). Second, deletion of Myomaker in satellite cells did indeed abrogate myonuclear accretion, and loss of myomaker and myonuclear accretion throughout the 8 week HIIT inhibited the mice's ability to perform and adapt to exercise stimulus (beginning at 3 weeks of HIIT). Third, based on measurements of Ankrd1/2 and Crsp3 the HIIT protocol resulted in transitory myofiber damage at 2 weeks of HIIT. Fourth, inhibition of satellite cell fusion for the entire 8 weeks of HIIT resulted in an increase in fibrosis. However, inhibition of satellite cell fusion beginning at 4 weeks of HIIT did not result in fibrosis or impair the ability of mice to exercise, but led to myofiber failure to hypertrophy. Fifth, even in the absence of satellite cell fusion to myofibers, myofibers are able to hypertrophy in response to myostatin inhibitors. Together these data suggest the following model. Satellite cells contribute to myofibers throughout the HIIT protocol, but only after 4 weeks of HIIT do myofibers hypertrophy. Based on the comparison of inhibition of satellite cell fusion throughout the 8 weeks of HIIT versus 4-8 weeks of HIIT, they conclude that satellite cell fusion during the first 4 weeks is necessary to prevent fibrosis and to ensure adaptation to exercise stimulus, while satellite cell fusion after 4 weeks enables myofiber hypertrophy. Thus they argue that the initial phase of satellite cell recruitment is to necessary to tolerate exercise stimulus, mitigate stress during the initial stages of HIIT, and prevent fibrosis, while the later phase of satellite cell recruitment is necessary for myofiber hypertrophy. Overall, this is a well-executed study that is clearly documented and explained and contributes to our knowledge of how satellite cells contribute to muscle hypertrophy.

1) I have only one substantive comment about the paper. The authors argue that the satellite cell contribution to myofibers during the first 4 weeks of HIIT serves a fundamentally different function than satellite cell contribution to myofibers during 4-8 weeks of HIIT. However, during the first 4 weeks of HIIT the mice were subjected to an increasing incline, while during the last 4 weeks the mice were subjected to increasing speed. Isn't the most parsimonious explanation of the difference in the function of satellite cell contribution during the first versus the second four weeks simply that these are different responses to increasing incline versus increasing speed? Would they see a difference if they simply increased monotonically either the incline or the speed throughout the 8 week HIIT? I do not propose that they re-do these experiments, but I think how to interpret the effects of inhibiting satellite cell fusion throughout the 8 weeks versus just the last 4 weeks of HIIT is unclear. At a minimum, the authors need to explicitly discuss this.

---

## [Author Response]

Summary:The manuscript from Goh et al. directly addresses a fundamental controversy on how muscle mass is built. More specifically if fusion of activated satellite cells (MP) are mandatory for hypertrophy. Reviewers felt that the approaches are unique and definitive. While there is excitement for the manuscript, there was a concern regarding the number of mice used in some of the experiments, During the discussion between the BRE and reviewers, it was felt that increasing the numbers of mice in these data sets would be desirable-if they are available. If not, more careful use of the statistics and more cautious conclusions.Below are the main points to consider. These points are given more context within the reviewers individual comments.1) The major weakness of the study is that the number of observations is low (down to 3), and some of the results therefore need to be interpreted with caution. Can more mice be included into specific experiments?

We increased the number of animals to at least n=4. Moreover, a post-hoc power analysis revealed that each significant difference detected is sufficiently powered (power of at least 80%).

2) The authors make conclusions regarding the phases (first 4 weeks versus 2nd 4 weeks) of HIIT, but there may be an alternative interpretation. Could these be different responses to increasing incline versus increasing speed?

This is an excellent point by the reviewers and one in which we have considered. Our Discussion has been amended to include this possibility.

3) The authors discuss muscle damage a lot, but the molecular data they provide (Figure 3—figure supplement 2) is not really on molecules related to damage or repair/regeneration: heat-shock proteins, embryonal myosin or creatine kinase would be beneficial.

We analyzed creatine kinase in the serum of WT and Myomaker^scKO 8w^ mice (+/- HIIT) as a marker of muscle injury/damage. We found evidence for muscle damage in WT mice after 2w and 4w of HIIT, and this was increased in Myomaker^scKO 8w^ mice. These data demonstrate that HIIT is associated with some level of muscle damage, which is exacerbated when fusion of muscle progenitors is blocked.

4) Clarification and caution in the Discussion.

We overhauled the Discussion as suggested.

Reviewer #1:[…] StatisticsThe authors use a mix of parametric and non-parametric testing, they need to make clear what the criteria for selecting each method was, and which p-values are derived from which method (for example in the legends).Is it possible to determine from 3 observations if they are normally distributed or not?Should non-parametric testing be accompanied with median and percentiles rather than mean an SD?There seems not that much difference in variance between groups, why was Welch test used?How about corrections for multiple comparisons?Would the time courses be better tested by using nested ANOVA, two-way ANOVA or a linear mixed analysis?

We thank the reviewer for these accurate criticisms and based on the comments we improved the statistics in the revised version of the manuscript. Through consultation with a statistician (Dr. Daniel Schnell, Cincinnati Children’s Hospital Medical Center) we made the following adjustments in the revised version:

a) In each of the figure legends we clearly indicated the statistical test performed.

b) The reviewer is correct that it is not accurate to determine if a data set is normally distributed from 3 observations. To deal with this issue we increased the majority of observations to 4, and did not perform a normality test on data sets with less than 4 observations. After this analysis of normality there is only one data set that required a non-parametric test (we used a Mann Whitney sum rank test for Figure 1G).

c) In the revised manuscript we used a 1-way ANOVA for data sets with one variable and a 2-way ANOVA for data sets with two variables. A multiple comparison test was used for each type of ANOVA.

ResultsThe authors discuss muscle damage a lot, but the molecular data they provide (Figure 3—figure supplement 2) is not really on molecules related to damage or repair/regeneration but rather to mechanical stress which might not always damage. If the authors still have extracts of muscles, they should be able to measure more relevant molecules such as heat-shock proteins or embryonal myosin (the latter can also be analysed on cryosections). It would also be useful if they could compare such markers between control and experimental groups, e.g. is the level higher in the mice that were tamoxifen injected at the onset than for other groups (controls and late injected.

We analyzed creatine kinase in the serum as a proxy for muscle damage. We did this in WT and Myomaker^scKO 8w^ mice at multiple time points after HIIT, and included the Results in the revised manuscript. We also now compare levels of mechanical stress genes between WT and Myomaker^scKO 8w^ mice. These data together show that HIIT-induced damage is increased in muscle that is fusion-incompetent (Myomaker^scKO 8w^ mice).

We did not observe evidence for embryonal myosin+ myofibers on cryosections. As another proxy for muscle damage/repair, we analyzed central nuclei in the myofibers after HIIT. Similar to the Myh3 analysis, the levels of central nuclei are low in the quadriceps (3-5% after 2 weeks of HIIT). The difficulty for detection of central nuclei or Myh3 in sections could be because they may occur in discrete regions of the muscle, and therefore missed by analyzing a section at the mid-belly of the muscle. In this instance, creatine kinase is likely a better readout for muscle damage because the levels are contributed from all muscles. Another reason to explain the low or absent levels of Myh3 and central nuclei is that there is some debate if those characteristics are associated with all types of muscle damage. This controversy was mentioned by reviewer #2. Since this issue is still being debated in the field, we prefer not to show the Myh3 or central nuclei data because we think the analysis needs to be deeper before making a concrete statement.

DiscussionAs mentioned, I find large parts of the Discussion problematic, for example:"Exercise-induced adaptations involve the response to myofiber damage, increased contractile output, and growth of the myofiber, but it has been difficult to ascertain if myonuclear accrual is required for early exercise-related injury alone, adaptive growth alone, or both."I have never seen anybody suggest that myonuclear accrual should not be related to repair of injury. Very unclear statement, rephrase.

We rephrased this statement.

"Surprisingly, the HIIT protocol described here uncoupled these adaptive processes."I am not sure the data really show this. You would need to explain better or delete.

We provide a better explanation of this idea in the Discussion.

"We observed an increase in myonuclear numbers at each stage of HIIT indicating that MP fusion occurs and that accrual is not a consequence of a single event."I am not sure I understand this. What single event? Do you mean onset of exercise? I would interpret the data to mean that there is a gradual increase in the number of nuclei with a somewhat lagging increase in CSA during exercise. Bruusgaard et al., 2010 showed by overload that the MP fusion seemed to precede the CSA growth, and thus hinting on a causal relationship. When the authors here state "It is interesting to consider that in WT mice myonuclear accretion occurs throughout HIIT but hypertrophy transpires at later stages." It sounds like a similar conclusion. Could the authors plot the data in a better way illustrating this point?

We agree that our data are consistent with Bruusgaard et al., 2010, and included this in the Discussion. One key advance in our study is that we not only show that increases in myonuclear numbers precedes hypertrophy, but also demonstrate that significant adaptive hypertrophy requires continued MP fusion and myonuclear accrual.

"This suggests that an increase in myonuclei does not necessarily indicate that hypertrophy is forthcoming, which could explain the inconsistencies of studies reporting a correlation between myonuclear number and myofiber size and other studies finding no correlation (Murach et al., 2018; Damas et al., 2018)."I don't understand this argument. I have not seen anybody suggest that an increase in myonuclei is in itself is sufficient to induce hypertrophy. As I understand it the general idea has been that more myonuclei might be required for hypertrophy, but that some hypertrophic signal (for example load) is also required. For example, inactive nuclei-rich fibres will also atrophy if denervated. I also don't understand how the findings of Murach et al., 2018 (is a review not a study) and Damas et al., 2018, could be explained in this way?

We removed this statement in the overhauled Discussion.

"Since our data indicate that evidence of myonuclear accretion cannot be used simply to predict myofiber size, we propose that addition of myonuclei may serve specialized roles depending on the stage of adaptation. For instance, myonuclei could be recruited to the myofiber during exercise initiation to allow adaptation to moderate damage whereas myonuclear accretion at the later stages of HIIT could facilitate adaptive hypertrophy. Collectively, these observations agree with reports in humans that resistance training-induced hypertrophy occurs after suppression of damage (Damas et al., 2016)".My interpretation would be that muscle fibres without new nuclei don't increase sufficiently in size and/or strength to cope with the forced HIIT and some of the fibres are ruptured. Muscle damage in untrained people that start to exercise abruptly and hard is well known (rhabdomyolysis), and here electroshock is used. I don't think this translates to specialized roles for nuclei depending on stage of adaption or that early accretion is necessarily related to damage. Has damage been observed in the control mice here? The title of Damas et al., 2016, rephrased here is misleading. As I understand it, the authors of that paper did not really suppress damage, they just compared synthesis during early phases with lots of repair and later phases with less repair and more radial growth. If this part is to be kept in the Discussion the relevance to the present paper needs to be explained better.

We now show damage in control HIIT-trained mice and have better explained our idea about myonuclear specialization while also discussing other possibilities. Specifically to the issue of damage, the creatine kinase data shows that injury begins early in HIIT and definitely precedes any increase in size, therefore we do not think that injury is solely a consequence of an inability to increase in size. It is possible the early increases in myonuclei lead to an increase in strength that help the muscle deal with HIIT, but strength was not measured early in the protocol, and we acknowledge that is a limitation.

Reviewer #2:[…] 1) I have only one substantive comment about the paper. The authors argue that the satellite cell contribution to myofibers during the first 4 weeks of HIIT serves a fundamentally different function than satellite cell contribution to myofibers during 4-8 weeks of HIIT. However, during the first 4 weeks of HIIT the mice were subjected to an increasing incline, while during the last 4 weeks the mice were subjected to increasing speed. Isn't the most parsimonious explanation of the difference in the function of satellite cell contribution during the first versus the second four weeks simply that these are different responses to increasing incline versus increasing speed? Would they see a difference if they simply increased monotonically either the incline or the speed throughout the 8 week HIIT? I do not propose that they re-do these experiments, but I think how to interpret the effects of inhibiting satellite cell fusion throughout the 8 weeks versus just the last 4 weeks of HIIT is unclear. At a minimum, the authors need to explicitly discuss this.

We agree that this alternative interpretation should be considered and included it in the Discussion.